# MELD-accelerated molecular dynamics help determine amyloid fibril structures

Bhanita Sharma [1] & Ken A. Dill [1,2,3✉]

It is challenging to determine the structures of protein fibrils such as amyloids. In principle, Molecular Dynamics (MD) modeling can aid experiments, but normal MD has been impractical for these large multi-molecules. Here, we show that MELD accelerated MD (MELD x MD) can give amyloid structures from limited data. Five long-chain fibril structures are accurately predicted from NMR and Solid State NMR (SSNMR) data. Ten short-chain fibril structures are accurately predicted from more limited restraints information derived from the knowledge of strand directions. Although the present study only tests against structure predictions – which are the most detailed form of validation currently available – the main promise of this physical approach is ultimately in going beyond structures to also give mechanical properties, conformational ensembles, and relative stabilities.

[1] Laufer Center for Physical and Quantitative Biology, Stony Brook University, Stony Brook, NY, USA. [2] Department of Physics and Astronomy, Stony Brook University, Stony Brook, NY, USA. [3] Departments of Chemistry and Physics, Stony Brook University, Stony Brook, NY, USA. ✉email: dill@laufercenter.org

Many different proteins can form amyloid fibrils, which are hallmarks of neurodegenerative diseases, such as Alzheimer's, Parkinson's, Huntington's, and spongiform encephalopathies,[1–6]. To better understand the mechanisms of amyloidosis, toward better mitigation strategies, it is essential to know how the protein molecules are structured in the amyloid assemblies. The experimental challenge is that, unlike soluble proteins, each amyloid fibril has multi-protein complexity and requires different conditions and procedures. Sometimes, this is surmounted by combining methodologies, such as solid-state NMR (SSNMR), cryo-electron microscopy (cryoEM), fiber diffraction, hydrogen-deuterium exchange, and electron paramagnetic resonance spectroscopy[7–9]. But, even this is often insufficient.

Computational modeling can complement experiments in determining amyloid structures[10–13]. Different methods, such as CYANA, XPLOR-NIH, CNS, UNIO, and Rosetta, have been used in conjunction with SSNMR data to determine amyloid structures[14–17]. However, these approaches also have limitations. Many computational methods are coarse-grained and simplified, often exploring only rigid molecular conformations, and requiring prior knowledge of loop locations or symmetries. Some methods are rooted in bioinformatics, drawing inferences from databases of existing structures, so they do not give populations, ensembles, or driving forces, which are ultimately needed to learn mechanical properties and relative stabilities of different conformations. In principle, these limitations can be circumvented by using atomistic molecular dynamics (MD) simulations, since they account for thermodynamic forces, atomistic details, and flexibilities, and preserve Boltzmann distributions of physical forcefields[18,19]. However, the rough energy landscape of protein intermolecular interaction and the longtime scales of fibrillization have typically precluded atomistic MD modeling.

Here, we describe a computational approach MELD (Modeling Employing Limited Data) for determining fibril structures. MELD accelerates MD simulations by using externally supplied information. MELD accelerated MD (MELD x MD) is what we call a *Bayesian* sub-haystack method because it utilizes Bayesian inferences and searches for the needle in the haystack, by dividing input constraint into different haystacks. MELD differs from other integrative modeling approaches in ways that are important here. First, its acceleration can be considerable, sometimes orders of magnitude, allowing for the handling of multi-peptide complexes. Second, MELD preserves a centrally important virtue of the physics that underlies MD—namely its ability to satisfy the Boltzmann distribution at equilibrium and Newton's Laws of dynamics. It gives populations, ensembles, and free energies, rather than just structures and rather than just single averages. Third, unlike methods of integrative modeling, MELD can utilize data that is vague, combinatoric, ambiguous, incomplete, or lacking in detail. MELD x MD has been validated in several venues. It has been successful in protein structure determination, including the blind competitive event critical assessment of structure prediction (CASP)[20–23]. MELD x MD was found in CASP 13 to be effective at giving protein structures from limited NMR data[24]. Here, we show that MELD x MD can assist in determining protein fibril structures, in conjunction with limited external data. For short chains, it correctly predicts 10 out of 12 fibril structures, from restraints derived from the external input information of parallel or antiparallel arrangement of strands. MELD also predicts five out of six longer-chain amyloid fibrils when given experimental NMR/ SSNMR data as input.

## Results and discussion

### MELD x MD predicts the structures of short-chain amyloid fibrils in 10 of 12 cases.

As validation, we first applied MELD x MD in predicting fibril structures of 12 short-peptide systems (fewer than ten residues per chain) (Table 1). The structures of these fibrils are already known from extensive experiments and are available in the database of 109 structures of Stanković et al.[25] (Supplementary Table 1). A note of caution is that we have used PDB as fibril structures, even though the oligomerization states could be different and are not known.

In order to achieve computational acceleration, MELD requires some form of guidance about the endstate, in this case the fibrils we aim to model. First, we need to assume the strand arrangements; since although all short fibrils have cross-$\beta$ structures, the arrangement of strands (parallel and antiparallel) and the patterns of connectivity among the monomers are different. Therefore, to reduce the conformational search space, we also assert whether the strands are parallel or antiparallel; see Table 1. We limit the conformational search by using the intermolecular distances to restrain the individual chains accordingly. The cross-$\beta$ sheet is characterized by a regular hydrogen-bonded distance between the constituent $\beta$-strands of 4.8 Å, along with approximate dihedral angle restraints for parallel (phi $= -119°$, psi $= +113°$) and antiparallel (phi $= -139°$, psi $= +135°$) strands. Therefore, we have incorporated inter-strand distance restraints between C$\alpha$ atoms of all corresponding residues and dihedral angle restraints to ensure an in-register alignment of the $\beta$-strands. The inter-sheet distances vary significantly (8.8–14.6 Å) for different amyloid fibrils. Since MELD can handle sparse, ambiguous, and uncertain information, therefore to generalize our protocol, we have applied inter-sheet distances of 10 Å between C$\alpha$ atoms of three central residues of each strand among one another (Supplementary Fig. 1). Our aim here has been to see whether MELD could produce correct structures with this generic protocol.

We found that MELD x MD successfully folded 10 of our 12 structures into fibril, starting from well-separated extended monomer chains (Fig. 1 a, b, c). To compute our prediction errors, we took the whole multi-protein aggregate as a single unit and calculated the C$\alpha$ root-mean-square deviation (RMSD) difference between the computed centroid of its most populated cluster (c0) relative to the reference biological assembly in the PDB. We superimposed the predicted and PDB structures by using the GDT_TS (global distance test, total score[26]), which represents the percentage of the structure within a certain cutoff to the native structure. The GDT_TS score has a value between 0–100. The higher the score, the better the quality of the structure. We also compute here the TM-score[27], a quantity in the range between 0 and 1, and which is independent of protein size. Usually, a score higher than 0.5 indicates that the two structures are similar. We also evaluated the percentage of native contacts by comparing the contact map of the reference PDB structure against our model fibril structure. Table 2 shows four structures (3fva, 3nhc, 3ow9, and 3ppd) having high similarity to the crystal structure. For 10 of the 12 short-chain fibrils, the most populated cluster from MELD x MD is within 3.5 Å backbone RMSD from the X-ray structure in the PDB.

### The side-chain orientations in short fibrils are well predicted.

Although amyloid fibrils are formed by different peptides having different sequences, the fibril structures have a basic steric-zipper pattern. Depending on whether the sheets are parallel or antiparallel, sheet packing pattern ("face-to-face" or "face-to-back"),

**Table 1 Our selection of fibril systems.**

**Short fibril systems (peptides smaller than ten resides)**

| PDB ID | Name | Sequence | Strand arrangement | No. of subunits |
|---|---|---|---|---|
| 2omq | Human insulin | VEALYL | Antiparallel | 8 |
| 2ona | Aβ(35–40) | MVGGVV | Antiparallel | 8 |
| 2onv | Aβ(37–42) | GGVVIA | Parallel | 4 |
| 3fva | Elk prion | NNQNTF | Parallel | 6 |
| 3loz | β2-microglobulin | LSFSKD | Antiparallel | 8 |
| 3nhc | Human prion protein (127-132) | GYMLGS | Antiparallel | 12 |
| 3nve | Syrian hamster prion (138-143) | MMHFGN | Antiparallel | 12 |
| 3ovl | Tau | VQIVYK | Parallel | 8 |
| 3ow9 | Aβ(16–21) | KLVFFA | Antiparallel | 12 |
| 3ppd | Prostatic acid phosphatase | GGVLVN | Parallel | 6 |
| 4onk | [Leu-5]-Enkephalin mutant | YVVFL | Antiparallel | 12 |
| 4r0p | Human lysozyme C (46–61) | IFQINS | Parallel | 10 |

**Long fibril systems**

| PDB ID | Name | Sequence length | Strand arrangement | No. of subunits |
|---|---|---|---|---|
| 2beg | Aβ(1–42) | 26 | Parallel | 5 |
| 2e8d | β2-microglobulin | 22 | Parallel | 4 |
| 2kj3 | HET-s(218–289) prion | 79 | Parallel | 3 |
| 2lnq | Aβ(15–40) Iowa mutant | 40 | Antiparallel | 8 |
| 2mxu | Aβ(1–42) | 32 | Parallel | 12 |
| 2m5n | Transthyretin(105–115) | 11 | Parallel | 16 |

Table shows the arrangement of strands (parallel or antiparallel), and the number of peptide subunits in the fibril structures.

and the orientation of sheets with respect to one another (parallel or antiparallel), amyloid fibrils are classified into eight classes[28].

In order to check the side-chain orientations relative to the PDB, we calculated RMSDs for all heavy atoms (Fig. 2). Since MELD incorporates flexibility, and since we are dealing with multiple chains, this flexibility affects the alignment of individual monomer chains for calculating RMSD. However, we observe that in most cases the side-chain orientations follow a similar pattern of steric-zipper interfaces as in the PDB. We did not restrain the side-chain atoms in our simulations. However, the restraints from dihedral angles and inter-strand and inter-sheet distances were sufficient to impose directionality to the sheet-sheet interface, which indirectly influences the side-chain orientations.

Now, in addition to comparing MELD predictions with experimental structures, we have also compared a few computational models. Even though, as far as we know, there are no comparable results on the short fibrils from other atomistic simulations, there are coarse-grained database-derived rigid molecule predictors. We compare here to Fibpredictor, Z-Dock, and ClusPro, which were previously tested against each other for all the eight classes of amyloid fibrils[29].

Fibpredictor uses statistical scoring functions combined with symmetry operations for β-sheet model building and replication to generate fibril structures, whereas Z-Dock and ClusPro perform protein–protein docking. Here we note that in the case of Z-Dock and ClusPro, backbone and side-chain configurations for the two sheets were taken from the original PDB as input for docking, whereas in MELD we have started from the random monomers to generate fibril structures. Fibpredictor

predicts the backbone and side-chain conformations on the fly during structure generation. In Fig. 3, we show the comparison of the lowest-free-energy MELD x MD structure with Fibpredictor, Z-Dock, and ClusPro. The RMSDs are calculated for all heavy atoms. The figure shows that MELD generates better structures than ClusPro in four cases; and better than Z-Dock in three cases. In one case (PDB 3ppd) MELD gives the lowest-RMSD structure with respect to the reference crystal among all others. These comparisons just show that physically parameterized MD with atomistic detail and flexibility are no worse at predicting structures than database-parameterized rigid-structure methods. Ultimately, the value of MD modeling is in other mechanistic and physical predictions, such as of polymorphs described below.

**MELD x MD also predicts the polymorphs in the few known cases.** Some amyloid fibrils are polymorphic, having multiple arrangements of the β-strands and side-chain packing[30,31]. Polymorphs are stabilized by hydrogen bonding, electrostatics, and aromatic stacking interactions. We observe that polymorphs appear in different MELD clusters.

Figure 4a shows the polymorphic structures generated in the MELD simulations for the VQIVYK sequence. In case of VQIVYK, four crystal structures can be found in the PDB (3ovl, 4np8, 2on9, and 5k7n). Our top MELD cluster (c0) resembles PDB 3ovl (RMSD 3.0 Å), whereas, cluster c3 and c12 resembles its experimental polymorphic structures (PDB 2on9 and 4np8). These polymorphic structures have different steric-zipper interfaces. The crystal structure of PDB 5k7n has only a

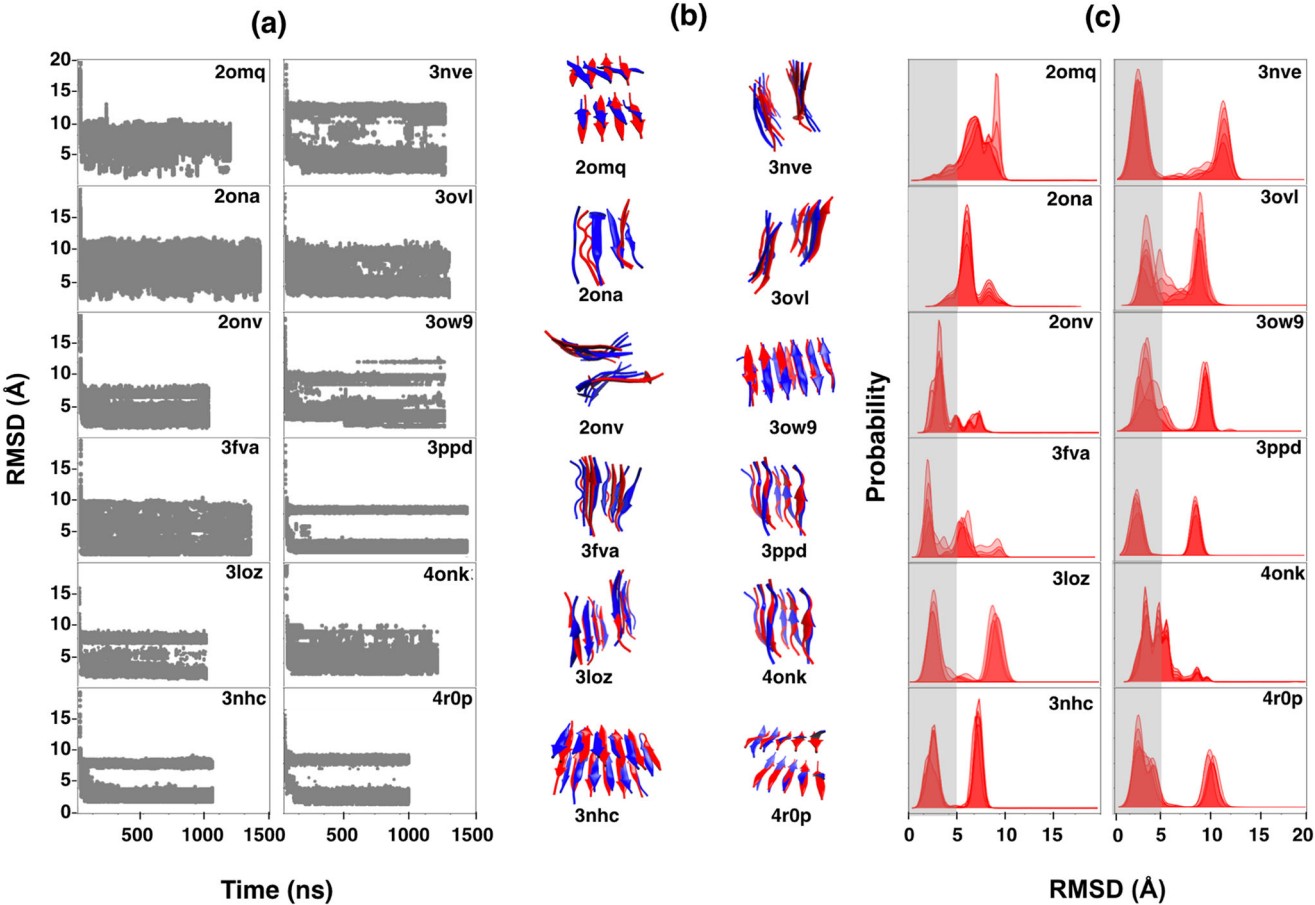

**Fig. 1 Computationally predicted vs. experimentally determined native structures of 12 short fibrils. a** RMSD errors for Cα atoms vs. time for the five lowest temperature replicas. **b** MELD predicted (blue) vs. PDB (red). **c** Population distribution of RMSD from native; gray regions show a 5.0 Å cutoff. MELD x MD obtains these with less than 1 µs sampling. The input restraints for distances and dihedral angles are derived from the external input knowledge of strand directions (parallel or antiparallel) of peptides in fibrils.

single parallel β-sheet, and therefore we excluded it from our analysis. We have also observed that the two most populated clusters (c0 and c1) of sequence NNQNTF are polymorphic structures (Fig. 4b) The top cluster of the NNQNTF sequence resembles PDB 3fva. The sheet organization (parallel/antiparallel) and inter-sheet distances of these two structures are the same, but these different structures result from different side-chain packings. Although the experimental polymorphic structure for NNQNTF is not available in the PDB database, the structure of the two steric-zipper polymorphic forms are discussed in various computational studies[32,33]. Here, we note that sequence MVGGVV also has two experimental polymorphic structures (PDB 2ona and 2okz). However, using our general restraint protocol for short fibrils, MELD failed to recover the polymorphic structures. On the other hand, the KLVFFA sequence also has three experimental polymorphic structures (PDB 3ow9, 2y2a, and 2y29). The top MELD cluster (c0) resembles PDB 3ow9. However, we excluded PDB 2y2a and 2y29 from our analysis, since they only have a single chain in the molecule's Biological Assembly file in the PDB.

Since the relative populations of clusters in MELD correspond to their free energies, we expect the more stable polymorphic forms to be the predominant populations. In these few cases, that's true. However, stability also depends strongly on environmental conditions such as pH, ionic strength, temperature, and polypeptide concentration, so we cannot yet draw firm conclusions about these predictions.

**Computing longer-chain fibrils, assisted by experimental data**. We now look at longer-chain fibrils. We considered six fibril systems that have PDB topologies measured by both solution NMR and SSNMR (Supplementary Table 1 and Supplementary Fig. 2). As background, we had already previously tested MELD x MD on predicting both individual native structures and protein–protein dimers[34–37]. Past experiences show that MELD success depends on the quality of the input data, and having proteins that are not too large, among many other factors. The present long-chain fibril systems present a size challenge for MELD x MD and depend on how many intramolecular and intermolecular restraints are imposed.

In our first test, we generated the fibril structures from the extended monomer chains at least 15 Å away from each other using experimentally based distance restraints and dihedral angle restraints. The oligomeric state (number of monomers) in the fibrils is the same as in the PDB. We observed that MELD x MD generates fibril structures below 5 Å for five fibrils except for PDB 2lnq. However, the convergence took longer for the larger fibrils (Fig. 5a). The convergence of 2lnq occurs above 500 ns but with sampling above 1 µs, the native structure is no longer populated in the top three clusters. Although the fibril size of 2lnq is smaller than 2kj3 and 2mxu, however, the number of restraints per amino acid is lowest for 2lnq.

In our second test, we started the simulation with fully extended well-separated chains to first form a trimer by applying experimental restraints (except for 2kj3) (Supplementary Fig. 3).

**Table 2 MELD x MD results for all fibril systems.**

**MELD x MD results for short fibril systems**

| PDB ID | RMSD (Å) | | | Population | | | % of native contact | TM-score | GDT_TS |
|--------|------|------|------|------|------|------|------|------|------|
| | c0 | c1 | c2 | c0 | c1 | c2 | c0 | c0 | c0 |
| 2omq | 7.2 | 8.0 | **5.8** | 0.3 | 0.2 | 0.1 | 43.61 | 0.33 | 48.13 |
| 2ona | 6.8 | 9.1 | **5.0** | 0.4 | 0.1 | 0.1 | 42.10 | 0.36 | 41.16 |
| 2onv | **3.4** | 4.2 | 4.4 | 0.4 | 0.2 | 0.1 | 45.33 | 0.50 | 58.23 |
| 3fva | **1.6** | 5.6 | 9.2 | 0.6 | 0.3 | 0.1 | 72.86 | 0.57 | 73.61 |
| 3loz | **2.3** | 8.5 | 5.4 | 0.3 | 0.2 | 0.1 | 49.31 | 0.52 | 63.54 |
| 3nhc | **1.9** | 7.5 | 2.0 | 0.3 | 0.2 | 0.1 | 62.50 | 0.74 | 70.83 |
| 3nve | **2.3** | 11.5 | 11.3 | 0.6 | 0.3 | 0.1 | 43.82 | 0.53 | 50.34 |
| 3ovl | **3.0** | 7.9 | 3.4 | 0.7 | 0.2 | 0.1 | 63.63 | 0.56 | 54.86 |
| 3ow9 | **1.8** | 9.5 | 2.8 | 0.4 | 0.3 | 0.1 | 63.41 | 0.76 | 72.92 |
| 3ppd | **1.0** | 8.0 | 3.3 | 0.4 | 0.3 | 0.1 | 95.95 | 0.78 | 89.58 |
| 4onk | **3.4** | 4.9 | 5.8 | 0.5 | 0.3 | 0.1 | 51.52 | 0.50 | 47.91 |
| 4r0p | **2.5** | 10.3 | 3.1 | 0.5 | 0.4 | 0.01 | 65.23 | 0.64 | 65.42 |

**MELD x MD results for long fibril systems**

| PDB ID | RMSD (Å) | | | Population | | | % of native contact | TM-score | GDT_TS |
|--------|------|------|------|------|------|------|------|------|------|
| | c0 | c1 | c2 | c0 | c1 | c2 | c0 | c0 | c0 |
| 2beg | 2.5 | **2.0** | 3.1 | 0.6 | 0.3 | 0.1 | 77.78 | 0.80 | 64.62 |
| 2e8d | **2.0** | 3.8 | 4.5 | 0.4 | 0.2 | 0.1 | 72.36 | 0.63 | 73.16 |
| 2kj3 | **3.2** | 3.5 | 5.3 | 0.4 | 0.2 | 0.1 | 55.36 | 0.50 | 54.16 |
| 2lnq | 5.2 | 5.9 | **4.8** | 0.3 | 0.1 | 0.1 | 52.64 | 0.51 | 50.34 |
| 2mxu | 3.4 | **2.9** | 4.5 | 0.5 | 0.3 | 0.1 | 76.06 | 0.85 | 60.87 |
| 2m5n | **3.0** | 5.2 | 4.3 | 0.4 | 0.2 | 0.1 | 53.95 | 0.76 | 56.11 |

MELD x MD results with RMSD to PDB reference and cluster populations for the top three clusters (c0–c2) are shown. The RMSDs are calculated for Cα atoms of residues. The cluster closest to native is shown in bold. The % of native contact, TM-Score, and GDT_TS score are also shown.

We then build up the final fibril structure from the trimer by adding more copies in again another MELD x MD simulation. This protocol converged much faster on the correct structures (Fig. 5b). The predicted long-chain fibrils in MELD are shown in Fig. 5c.

The reason for selecting the trimer structure as a building block for generating fibrils instead of the dimer (or monomer) is because the dimer structures generated in MELD are different than the fibrils, due to high intra-monomer β-association propensity between residues within individual peptide monomers. Whereas, the trimeric structures are much closer to the fibril reference. However, in case of fibril PDB 2kj3 (chain length 79 amino acids) cross-β-sheets are formed within the individual chains and also with other chains. And since it is longer in size-length, therefore, we first generated the monomer structure from the fully-extended chain by applying experimental intra-monomer distance restraints and dihedral angle restraints; which then converted to the final fibril structure using inter-monomer distance restraint data. All the predicted structures are shown in Table 2. The predictions are close to the native fibril structures, as also indicated by (i) the GDT_TS score, (ii) the TM-score, and (iii) the percentage of native contacts. In most cases, the final states are similar for either of the two protocols above, but starting from trimers converges faster.

**What is the minimal information needed for predicting fibrils?**
In the simulations above, we have used information that would be readily available for structure prediction. But, could we have used less? Here, we simulated a set of systems by removing information step by step from the general MELD protocol. For short fibrils, these are System S1 (without dihedral angle restraint data),

system S2 (without dihedral angle and with limited intra-monomer restraints), and system S3 (unguided MELD simulations) (See Supplementary Figs. 4–6). For long fibrils, we chose four fibrils 2beg, 2e8d, 2mxu, and 2m5n. The different sets of simulations are System L1 (without dihedral angle restraint data), System L2 (restraint protocol based on knowledge of strand arrangement), and System L3 (only inter-strand distance restraint of 4.8 Å between parallel strands).

The results of all the predicted structures of these different systems are shown in Supplementary Figs. 7–12 and Supplementary Tables 2–3. We observed that when all the distance restraints information were used, the quality of predicted structures are still good in all cases even after the removal of dihedral angle restrain (systems S1 and L1). The ratio of populations of different clusters also does not change much. For systems with limited distance restraints data (system S2 and L2), the most populated cluster is within 5.0 Å in 7 out of 12 cases for system S2; and 3 out of 4 cases for system L2. These results suggest that while extensive experimental data may naturally give good predictions, however, a limited number of qualitatively informative restraints can still generate correct structures. The unguided MELD simulations (system S3 and N3) generates structures with large Cα RMSD errors. However, even then, in case of short fibrils (system S3), for 8 cases out of the 12, the most populated cluster gives strand arrangements (parallel/antiparallel) the same as in the experimental structures. For the other four fibrils, the oligomeric structures are found to be a random mix of both parallel and antiparallel strands.

Figure 6a shows that for short fibrils, restraints per residue above 0.5 mostly lead to successful prediction. In Fig. 6b, c, we observe that the quality of MELD prediction for longer fibrils depends on both the number of restraints imposed and the size of

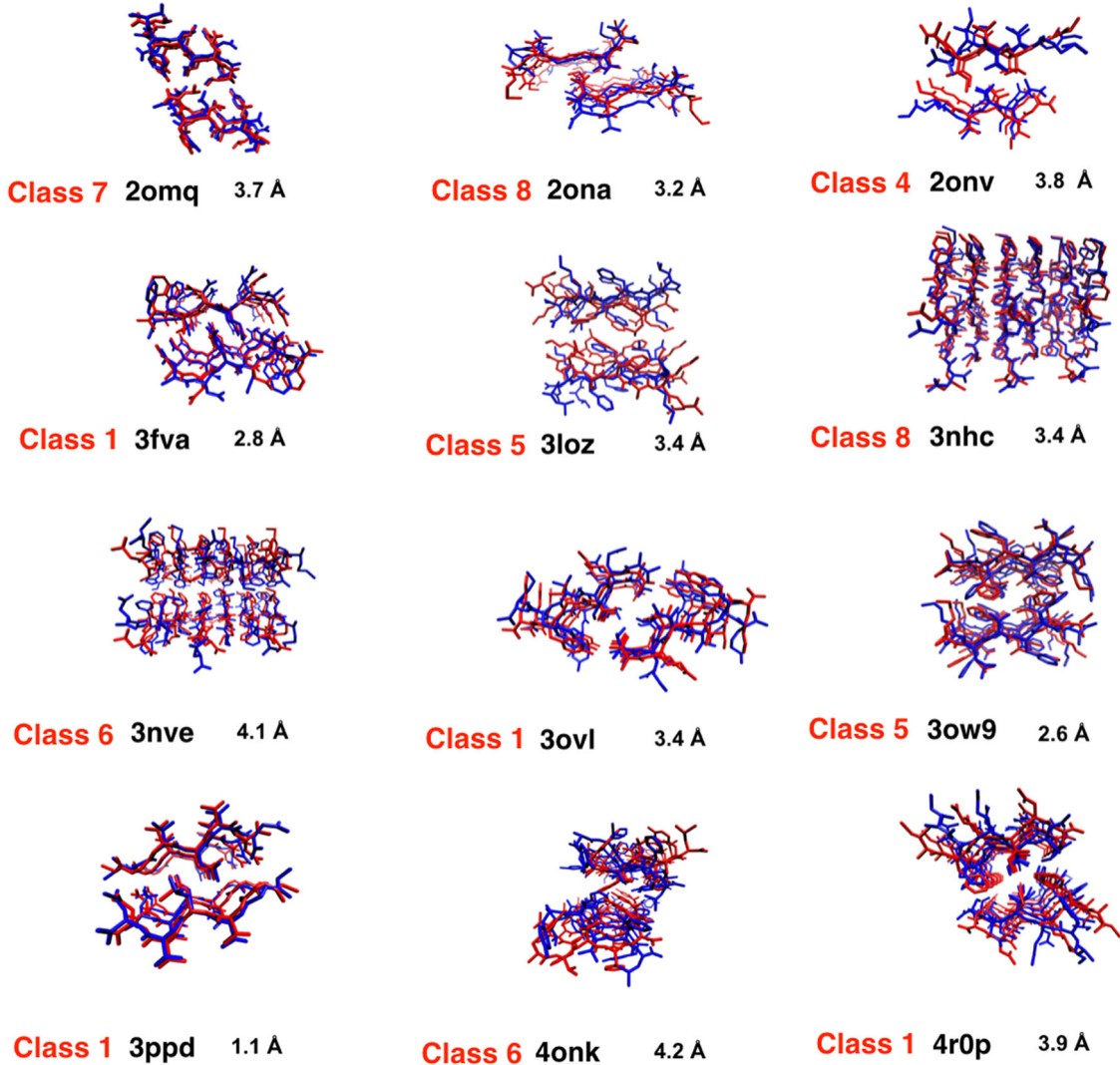

**Fig. 2 Side-chain orientations predicted for lowest-free-energy MELD structure.** (Blue) MELD x MD predicted structures. (Red) PDB structures. For PDB 2omq and 2ona, the MELD structure from the cluster closer to the native are considered. MELD correctly predicted steric-zipper interface patterns in most cases. The RMSDs are calculated for all heavy atoms.

the fibril. For fibrils of larger size, the prediction quality heavily deteriorates in absence of a sufficient amount of qualitatively informative restraints. The restraints per residue used for different systems are shown in Supplementary Tables 4, 5.

**The failure modes, and recovering from them**. When our predictions fail, it can be due either to force field errors or to insufficient sampling by the MELD x MD strategy. Most of our failures are from system S3 and L3, where the restraints were least informative. However, even with experimental SSNMR restraint data for long fibrils, the RMSD deviation for the top MELD cluster for PDB 2lnq are higher. On the other hand, with our general restraint protocol for short fibrils, we failed to get the correct structures for PDB 2omq and 2ona.

We compared our MELD x MD models to a pure MD simulation that was initiated from the experimentally determined PDB structure and converged (Supplementary Fig. 13a). We observe that for PDB 2omq, the native fibril is not stable in the force field. On the other hand, the failure of PDB 2ona is due to the failure of our inter-sheet distance restraint protocol. The failure of PDB 2lnq is mostly associated with the intrinsic

flexibility of the peptide in simulations, as the number of inter-monomer β-sheet alignment restraints were few. The restraints per residue are also lowest for 2lnq among all the longer fibrils. In order to recover the failed MELD structures, we carried out another set of simulations adding a few inter-monomer β-sheet alignment restraints of 4.8 Å between the corresponding residues for PDB 2lnq. In case of PDB 2ona and 2omq, we added some accurate inter-sheet distance restrains according to the reference PDB structures in MELD. This improved the structure prediction with a lower RMSD top cluster in MELD (Supplementary Fig. 13b).

We can determine when the method is expected to be successful by looking at the populations it gives of the MELD clusters. When a cluster has a high population, it implies it has the lowest free energy, so, according to the model, this should be the best prediction. This is what we find. In most cases, the clusters with populations higher than 40%, are the correct structures. When the population of the first cluster is high, the centroid structure of the most populated cluster accurately represents the fibril structure. Here we note that the number of restraints per residue among larger fibrils is smallest for 2lnq; and the population of the top cluster is 30%, whereas for other

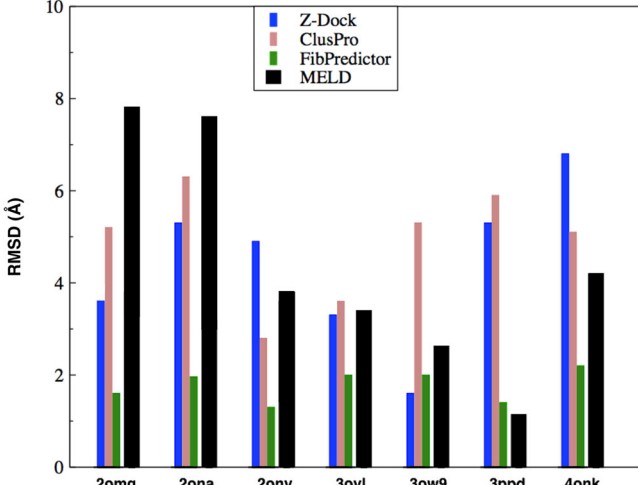

**Fig. 3 Comparing MELD to empirical structure predictors; Fibpredictor, Z-DOCK, and ClusPro, for short fibrils.** The RMSDs are calculated for of all heavy atoms. This histogram shows that modeling of the full physics, flexibility, and solvation, with the attendant advantages for ultimately understanding mechanisms and motions,- gives native structures of about the same accuracy as data-derived predictors. This is a useful validation since native structures are the main experimental evidence having such detail and precision.

structures are 40% or higher. When a simulation gives the best population below 20%, it usually signals unsuccessful predictions. Most of these predictions are for systems S3 and L3, where the least informative restraints were used (Fig. 7).

## Conclusions

A goal in computational biology is to model biomolecules with physics-based MD methods because, in principle, they can go beyond just static structures and also predict conformational populations and free energies, motions, and biologically important actions. Even so, predicting native structures is a useful milepost test for MD modeling because data is so extensive and granular. Modeling amyloid fibrils has previously been challenging because they are multi-molecular and large. Here, we show that using the MELD method for accelerating MD can be useful for this. For short-chain amyloids, MELD x MD correctly predicts 10 out of 12 fibril structures, from restraints information derived from the knowledge of parallel or antiparallel arrangement of strands. In most cases, the side chains are found to form steric zippers having fairly accurate side-chain orientations. MELD also correctly predicts five out of six longer-chain amyloid fibrils when given experimental NMR/SSNMR data as input. While we test here only structure predictions, the ultimate value of MD modeling is in giving Boltzmann populations and free energies; dynamics, motions, and flexibilities; and giving transferability

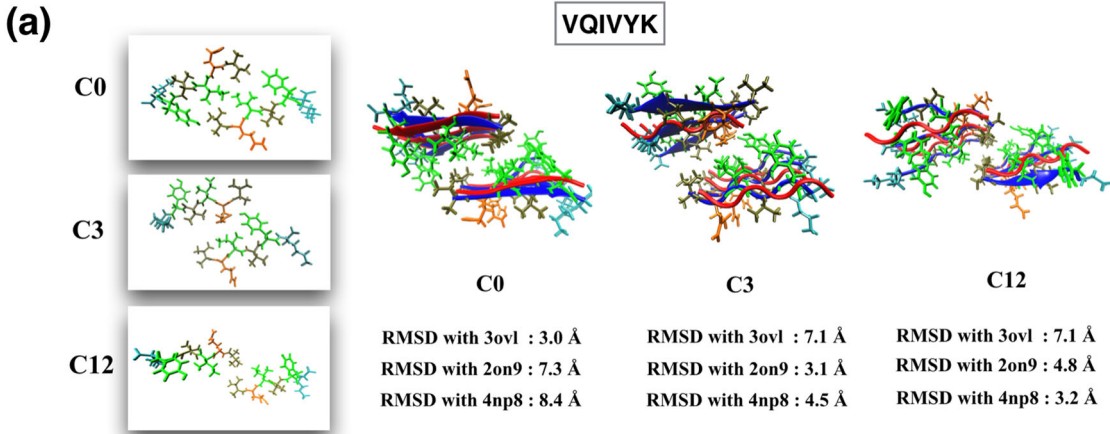

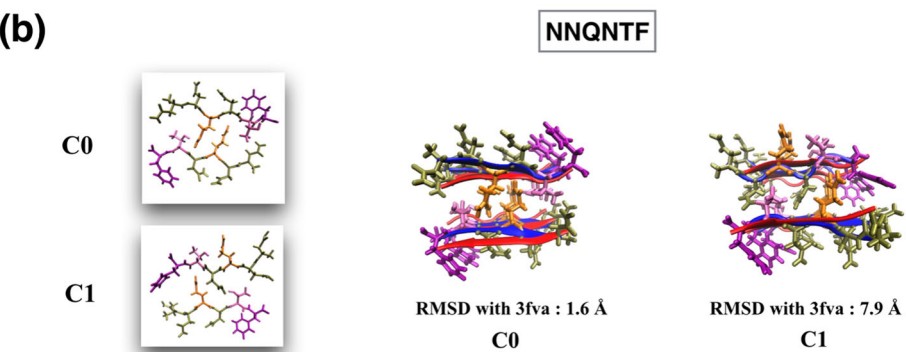

**Fig. 4 Polymorphs of VQIVYK and NNQNTF sequences. a** Polymorphic structures of VQIVYK sequence. **b** Polymorphic structures of NNQNTF sequence. The crystal structures are shown with $\beta$ strands in red; and MELD x MD $\beta$ strands in blue. Side-chain conformations are from MELD x MD predictions (coloring according to residue names in VMD). RMSD errors are calculated for C$\alpha$ atoms of residues.

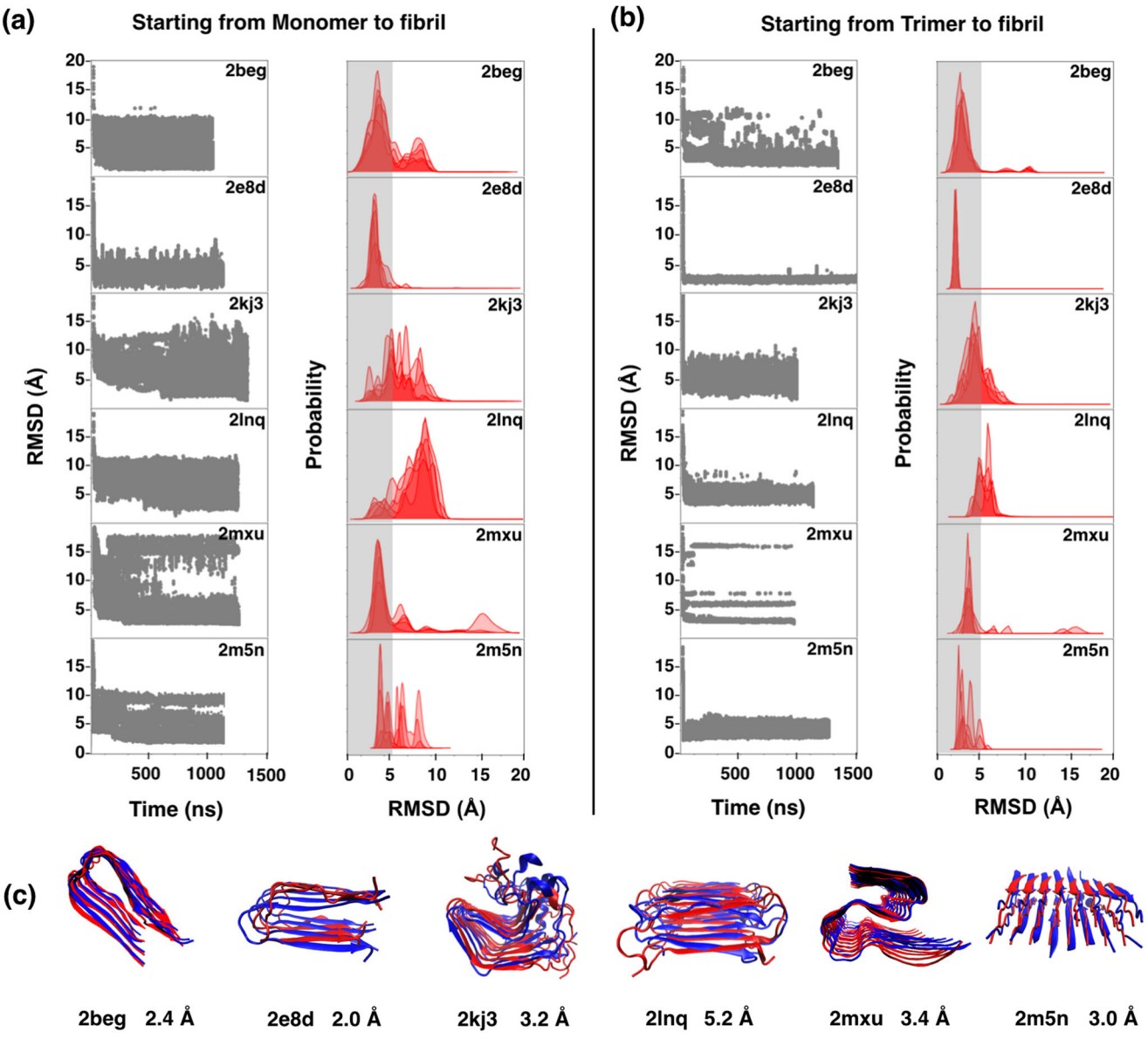

**Fig. 5 Computationally predicted vs. experimentally determined native structures of six long fibrils. a** starting from extended monomers to fibril and **b** starting from trimer to fibril structure. **c** MELD predicted (blue) vs. PDB (red) for long fibrils. **a, b** show the RMSD vs. time and populations distributions of the long-chain fibrils. RMSDs are calculated for Cα atoms of residues. The RMSD data are for the five lowest temperature replicas, the same replicas that were clustered for analysis. The error bar is shown at 5.0 Å, the cutoff used to define a native fibril. For our test simulations of generating fibril from random monomers, convergence took a much longer time; while in the case of simulations starting with trimers, convergence is much faster to the native fibril structures.

among different systems. We also show that MELD correctly predicts the polymorphic structures for two known polymorphic sequences. In general, MELD x MD may be a useful tool for broadly studying the physical properties of amyloid fibrils.

## Methods

**How MELD leverages data**. MELD can accelerate MD simulations of proteins by incorporating external information. It uses Bayesian inference,

$$p(x|D) = \frac{p(D|x)p(x)}{p(D)} \sim p(D|x)p(x) \qquad (1)$$

where $p(x)$, the prior probability, is the Boltzmann probability distribution of the MD force field in the absence of any external data. $p(D|x)$, the likelihood function, is the probability of the externally supplied data (D) given the structure; it is proportional to $e^{-E_c(x)/kT}$, where T is the temperature and $E_c(x)$ is the overall constraint energy as formulated by MELD. p(D) is a normalization factor and cancels out when considering ratios of the posterior probability of sampling

different configurations. MELD is a tool that speeds up MD modeling while preserving proper Boltzmann populations when given external knowledge of some target objective; this can come from a wide variety of sources, including experimental data, or physical insights, or homology models, etc. MELD can handle information that is sparse, ambiguous, or uncertain. Sparse refers to accurate but insufficient data to specify a structure. Ambiguous denotes data that are not very precise. Uncertain refers to data that are only partially correct, with a subset of wrong information that would lead to incorrect structures. To avoid kinetic traps MELD x MD uses Hamiltonian-Temperature Replica Exchange Molecular Dynamics (H,T-REMD)[38,39]. The top replicas have a high temperature and weak restraints, allowing the system to sample for a large energy landscape. Conversely, at low replicas, the restraints are strong and temperature is low, therefore focusing sampling on relatively funneled regions. We identify the low-energy conformations by clustering populations. For more details of MELD, see references (20, 21).

**MD simulation parameters**. All our MELD x MD simulations are carried out using a version of the OpenMM[40] and run on graphical processor units (GPUs). Each simulations are performed with 28 or 30 replicas. The temperature in replicas increases geometrically from 300 K in the lowest replica to 450 K in the highest.

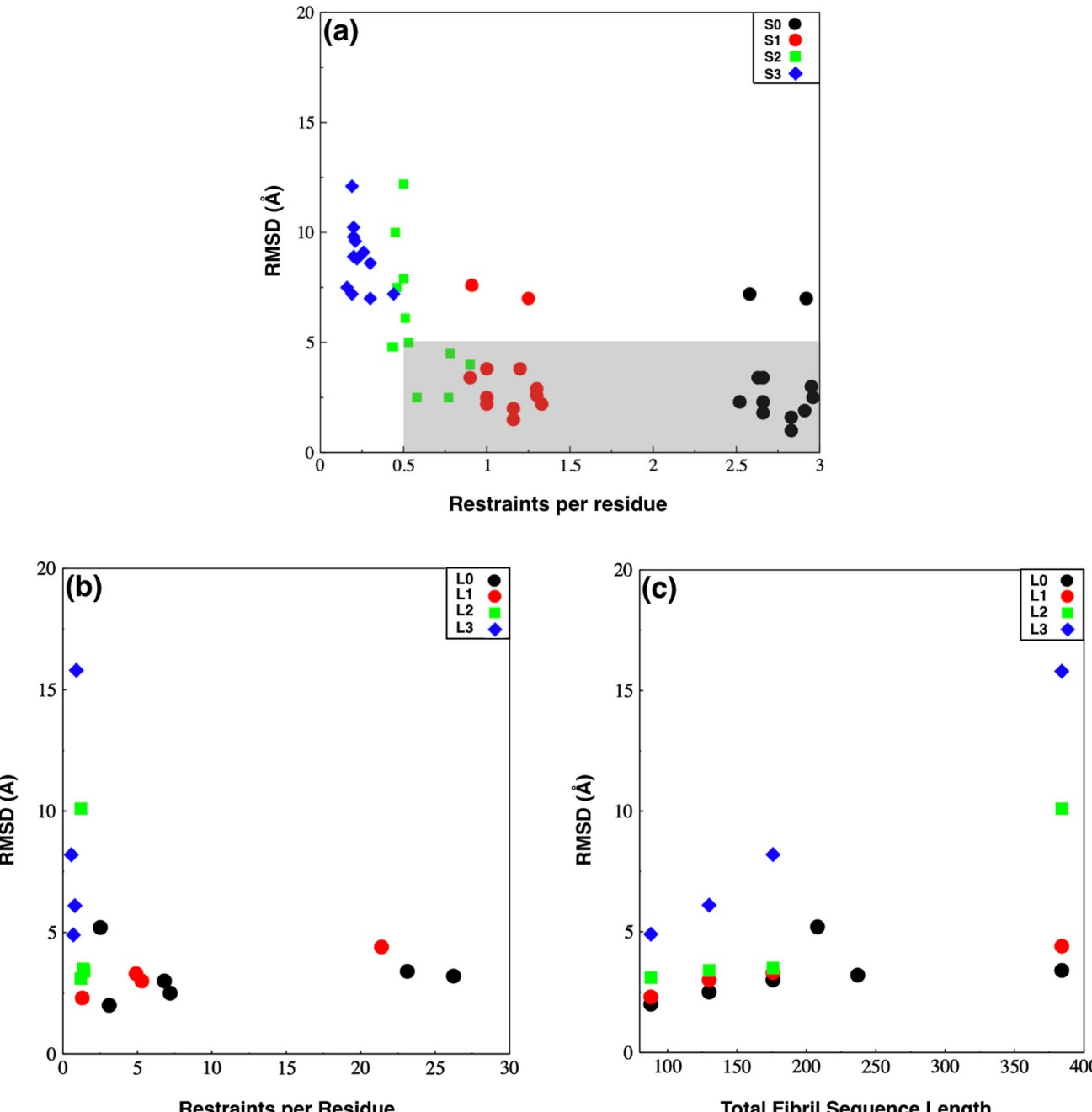

**Fig. 6 Prediction errors with limited restraints. a** Prediction error as a function of the number of restraints imposed per residue for short fibrils with limited restraints. System S0 is a reference for our general restraint protocol for short fibrils. In System S1, dihedral angle restraints are removed. In S2, distance restraints are limited. And in S3, only a single set of distance restraints between central residues of each peptide strand are imposed. Predictions are mostly successful with restraints per residue above 0.5 (gray region). **b, c** Prediction error for long fibrils; **b** as a function of the number of restraints imposed per residue, and **c** as a function of the fibril length. System L0 is a reference for when all NMR restraints are imposed. In system L1, dihedral angle restraints are removed. In system L2, the restraint protocol is similar to short fibrils. In system L3, only inter-monomer distance restraints of 4.8 Å for parallel strands are imposed.

The restraint force constantly weakens climbing above the replica ladder. At low replica index, force constant is strong (250 kJ/mol/nm$^2$) and becomes zero at high replica index, changing exponentially from the lowest to the highest replica. The exchanges between replicas are attempted every 50 ps, and the acceptance probability of replica exchanges are typically at about 30–50% (Supplementary Fig. 14). In some cases when the exchanges between the replicas were poor, increasing the number of replicas to 60, improved the exchanges (Supplementary Fig. 15). We use AMBER ff14SB-side force field[41] and GB-neck2 implicit-solvent model (igb = 8)[42]. Simulations are run with steps of 4.5 using a Langevin integrator with a friction coefficient of 1.0 ps$^{-1}$. Hydrogen masses are adjusted to 4.0 Da, keeping the heavy atom and hydrogen pair mass the same. Our typical inputs to MELD are (i) the

initial configuration of peptides that are randomly placed, (ii) externally supplied distance restraints information, and (iii) externally supplied dihedral angle information. Of all possible restraints generated, only a fraction are enforced at each time step; the energies of the restraints are calculated at each exchange step, and the lowest-energy restraints in each replica are activated. For short fibrils, restraints are derived from the external input knowledge of the parallel/antiparallel arrangement of strands. In Supplementary Fig. 2, we have shown the pairwise contacts of distance restraints used in our simulations for long fibrils. Among all structures, 2mxu has the highest restraint data, however, 2kj3 contained the highest number of nonlocal restraint data. Restraints per residue are also highest for 2kj3, while for 2lnq the lowest. We generate around 1-μs trajectories for the systems. The

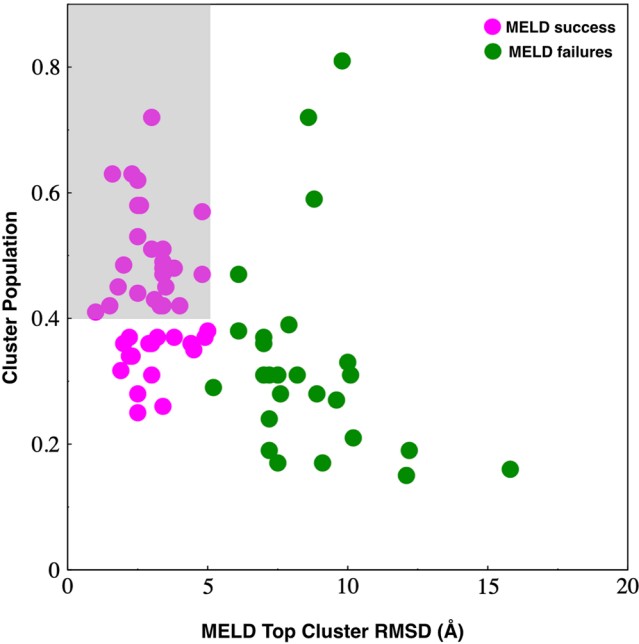

**Fig. 7 MELD top cluster population and MELD success.** When MELD converges to a dominant population, that's usually the successful prediction. In most cases when MELD populates a single structural ensemble more than 40%, it identifies the native structure (to within 5.0 Å RMSD, gray region). In contrast, most predictions below 20% are for systems S3 and L3, for which no informative restraints are used.

convergence of our replica exchange simulations are checked with RMSD histograms of all the replicas relative to the last frame of the simulation. Converged simulations give overlapping histograms. It is observed that in some cases the higher replica index trajectories are not converged (Supplementary Fig. 16).

**Ensemble processing**. We extracted our results by clustering our trajectories using an average-linkage hierarchical agglomerative algorithm with a $\epsilon$ value of 2 Å, with a combination of scripts included with MELD x MD and CPPTRAJ[43] from AmberTools17. Trajectories from the five lowest temperature replicas are combined, and the first 250 ns of trajectory frames were rejected from clustering, considering the equilibration period for the systems. As a distance matrix between structures, we used the RMSD of Cα atoms. To check the side-chain orientations of short fibrils relative to the reference PDB, we calculated RMSDs considering all heavy atoms. The centroid of different clusters represents different conformations according to cluster population and we have selected the three most populous structures for analysis. We arbitrarily define a threshold in which structures within 5.0 Å RMSD from the reference PDB as a successful prediction.

**Forcefield stability tests**. To check the stability of all fibrils in our force field and solvent model, we have carried out single-trajectory MD simulations, starting from the PDB structure of the fibrils. The ff14SBside protein force field was used with the GBneck2 implicit solvent, same as in MELD x MD. The systems were first energy minimized for 10,000 steps with 5000 steps of steepest descent minimization, followed by 5000 steps of conjugate gradient minimization. All simulations were run for 100 ns of production at 300 K. We consider a fibril structure as stable if the average RMSD was below 5 Å.

**Statistics and reproducibility**. MELD x MD simulations with different randomly placed initial configurations of peptides show that, in the limit of convergence sampling, using the same restraints information reproduces the same lowest free energy structures. No data exclusion was performed, and no blinding methods were used in data analysis.

**Reporting summary**. Further information on research design is available in the Nature Research Reporting Summary linked to this article.

## Data availability
The datasets analysed in this "Article file" are provided with this paper and in the "Supplementary Data" file. The remaining datasets used in the "Supplementary Information" are available from the corresponding author on request.

## Code availability
MELD is freely available to download from GitHub at https://github.com/maccallumlab/meld.

The version of the code we used in this study is available at https://github.com/maccallumlab/meld/releases/tag/0.4.14 or https://doi.org/10.5281/zenodo.5083678[44].

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

## Acknowledgements

The authors are grateful to Christopher Jaroniec for suggesting the problem to us. This work has been supported by National Institute of General Medical Sciences 5R01GM12581303 "MELD: Accelerating MD Modeling of Proteins Using Bayesian Inference", National Science Foundation ACI1713695 "Petascale Integrative Approaches to Protein Structure Prediction" and Fontreta LRAC allocation MCB20004 "Petascale Integrative Approaches for de novo Protein Structure Prediction". We appreciate the Laufer Center for Physical and Quantitative Biology, Stony Brook University for support. We are also grateful for the support of the Frontera and Blue Waters sustained-petascale computing project. The authors thank Emiliano Brini, Cong Liu, Roy Nassar, and Alberto Perez for insightful discussions.

## Author contributions

B.S. planned and analyzed the results and wrote the paper; K.A.D. supervised the research and coauthored the paper.

## Competing interests

The authors declare no competing interests.
