## [Peer Review File · Communications Biology]

Reviewers' comments:

Reviewer #1 (Remarks to the Author):

Previously the authors' laboratory has developed an approach that applies Bayesian inference to molecular dynamics simulations which has shown efficacy in the prediction of protein structures using limited and imperfect information. The acceleration this method provides increases the tractability of applying physics-based guided molecular dynamics simulations to problems otherwise typically solved using knowledge-based molecular modelling packages. In this manuscript the authors make use of this technique to build models of amyloid fibrils.

While the amyloidogenic peptides appear to represent a valid target for the MELDxMD method the paper in its present form requires significant changes before it would be suitable for publication. In particular, the application of this technique to the target molecules considered is not sufficiently systematic to really really validate its use. In addition more consideration should be given to the amount of extraneous information provided on the quality of the resulting models. Referencing needs to be improved and key details are absent from the methods section. I provide details below.

Major points:

1. For the short-chain fibrils, the amount of information being provided for a relatively simple system is extensive (peptide backbone restraints; inter-strand distances; inter-sheet distances; parallel/anti-parallel arrangement). With such constraints it is unsurprising that the arrangements that emerge are consistent with the PDB structures, and given the significant correlation between side-chain directionality and backbone dihedrals (which have been constrained) it is not especially informative that side-chain orientations are similar between the models and the structures. It is of greater interest as to how the correspondence with the reference structure degrades with the removal of each piece of information for the different peptides. Essentially the purpose of this manuscript is to advocate use of this software in the modelling of fibrils, so a proper go at validating of the minimal requirements in this particular application is crucial.
2. Contacts are designated as Cbeta atoms within 7.5Å of one another. Why this definition rather than consideration of the side-chain atoms - the reason should be stated.
3. The authors compare with modelling one software application that apparently applies rigid rotation and translation to build fibrils with model optimisation limited to side-chain rotamer packing; and two that are based on docking. There is no mention in the methods or elsewhere of the input parameters used, and it is not explicitly stated whether the same a priori information is used as input - all of these details are essential and must be stated in the methods section. It is surprising that a comparison has not been made with a software that allow similar degrees of freedom of molecular dynamics: a more informative comparison would involve use of molecular modelling software e.g Rosetta or Modeller that is provided with the same input information through the use of dihedral and ambiguous distance constraints.
4. In regards to the identification of polymorphs, is there any independent experimental evidence you can provide for any of these peptides that support the relative populations you are observing? This would give greater confidence that they are not merely computational artefacts. If there genuinely is no such information then you should refer to prior examples where the ensembles identified by the sof
5. A more systematic approach to address the question posed by the authors "how much information does MELDxMD need" really requires consideration of several structures. n=1 just isn't enough to give confidence that the requirements have truly been tested

6. In the limitations section, the standard MD of the 2mxu structure is said to deviate from that seen in the PDB. Were the same dihedral/distance restraints used in the MD simulation as were used with the MELDxMD approach? If not, these cannot really be directly compared

7. Referencing is poor in places and needs to be improved

- 14,15,16,17 in paragraph 1 is misleading, these references do not actually provide examples of ssNMR being used to model amyloids. References should be to studies that actually did this.
- 27,28,29,30 on p3 are not examples of simulations leading to kinetically trapped oligomeric structures, I cannot see why they are included here
- Long enumerated lists of exemplary references to make a single point is lazy referencing and can be substituted with one or two particularly relevant papers instead
- (18,19,20,21,22,23,24,25) is excessive self-citation and can be trimmed to the most pertinent papers
- Look out for any other statements where apparently supporting references are provided that are actually not

Minor points:

1. The term 'sub-haystacking' on p2 is a very obscure one. This requires some exposition or substitution with another phrase.

2. It should be explicitly stated in the text and/or supplementary table 1 and table 2 that the information in the column 'strand arrangement' provides information that is used to direct the simulations

3. Supp tables 1 and 2 column 2 heading Name(Å) appears to be a typo

4. The PDB structure 2kib is the only short peptide dataset that was not solved by X-ray crystallography but by refinement in XPLOR based on SSNMR data. As such is itself a model built from constraints; it could be removed from this test set or this characteristic at least acknowledged

5. Provide units on Fig. 2 Y axis

6. p5 restraints enforced at 80%... 80% of what? Their magnitude?

Reviewer #2 (Remarks to the Author):

In this paper by B. Sharma and K.A. Dill, the authors apply Modeling Employing Limited Data accelerated molecular dynamics (MELD x MD) to provide atomistic computational models of a set of amyloid-forming protein fibrils, both "short-chain" (13) and "long chain" (6), using limited experimental information. Moreover, because MELD x MD yields not just single structures but conformational ensembles, the authors claim that their method can also generate polymorphs with similar backbone conformations but distinct side chain packing.

Determining atomic level structures of amyloid fibrils, essential to understand many human diseases, is challenging. Therefore, novel computational approaches, as it is the case of MELD x MD, are welcome. I expect the results of the present work would be beneficial for the community of structural biologists, protein folding and beyond.

In my opinion, the author's claims are not convincingly supported by their data and the clarity of their explanations could be improved:

Major comments

1) The choice of the experimental structures should be better explained.

The known limitations of MELD x MD i.e., short sequence lengths, preference for monomeric and dimeric assemblies, low net charge, high secondary structure content, etc. should be mentioned explicitly and the authors should clearly state whether these limitations of MELD x MD were taken into account for the selection process. For the long chains, the authors mentioned that they were arbitrarily chosen. Is this actually the case or did they try to test different primary structures, oligomeric states and strand arrangements?

2) The number of chains in the biological assembly (oligomeric state) is an overlooked restraint. The authors emphasize that their method can predict accurate structures with limited experimental data (apart from the obvious knowledge about the sequence) but the number of subunits in the fibrils is barely mentioned. I feel the assumed oligomerization is a very important restraint that should be better explained and discussed. I recommend the authors to include the presumed oligomerization state in Tables 1 and 2 (the authors should be aware that they took the biological assembly file from the PDB but the true oligomerization state of a protein is indeed difficult to ascertain and could be different).

Moreover, because MELD x MD has so far only been able to deal with monomers or dimers, the authors should explain clearly why now they can model higher order oligomers (tetramers, octamers and so on).

I would remove all mentions in the text along the lines of “using only knowledge of strand arrangements”, since this is strictly speaking not true. What I mean is to remove words like “only” or “even” that tend to over-emphasize their claims.

3) The precise experimental data that was used to model the 6 long chain fibrils is not clear. The structures of the chosen long chain fibrils shown in Table 2 have been solved using different experimental methods (solution NMR, solid-state NMR and others) but it seems that the authors used only NMR distance restraints and backbone restraints. They should specify which and how many restraints per amino acid were employed. Then they could delineate how much information does MELD x MD actually need for predicting long chain fibrils.

Because pdb 2beg was solved by solution NMR, where the number of restraints per amino acids is typically much larger than for solid-state NMR, one would expect a much lower RMSD for 2beg (Figure 5) but this is not the case. Can the authors comment on that? In the main text and in the legend of figure 5, it should be mentioned that for 2beg the authors used solution NMR data.

Related to this issue, are the differences seen in Figure 5 statistically significant? In other words, would the authors recommend to use specific NMR restraints to guide MELD x MD simulations or generic restraints are enough? Such important conclusion of the author’s results is not clearly stated in the text.

4) Polymorphs are only generated for one fibril.

The claim that MELD x MD can generate polymorphs would be more robust if they could repeat their analyses for other fibrils. Then it would also become clearer whether the crystal structure is always the most stable polymorph.

Minor comments:

5) After reading the introduction, one would erroneously conclude that atomistic MD simulations of protein fibrils have not been attempted. The authors should cite some key references e.g. <https://doi.org/10.1038/srep33156> and <https://doi.org/10.1021/acs.jpcb.5b11380>.

6) On page 1, the sentence “And, computational methods are often rooted in bioinformatics, ...” is very hard to follow. Please consider rewriting.

7) Page 2, Because this work is based on the MELD x MD method, references 18-25, which represent the state-of-the-art of MELD x MD, should be better explained.

8) Page 2, I believe MELD x MD was employed in CASP11 instead (reference 21).

9) The arbitrary distinction between short- and long-chain fibrils (e.g. 10 residues threshold) should be briefly mentioned.

10) Please check for consistency between RMSDs definitions, sometimes is C α only, others C α and C β , backbone, heavy atoms, etc. It is mentioned that the RMSD is calculated relative to the first model in the PDB but sometimes there are 10 experimental models so other approaches (e.g. computing their centroids) might be more accurate. The authors should justify their chosen method.

11) Page 2, please add references for the scoring functions (GDT_TS, Tm-score, etc.)

- 12) On page 4 it is mentioned that "The side-chain orientations are well predicted from the backbone restraints alone". How is this quantified?
- 13) On page 4, I would move the paragraph starting with "All these methods" to the discussion section since this is more a conclusion than a result.
- 14) On page 5 they claim that "MELD X MD successfully predicts the fibril structures starting from extended monomers". Do the simulations of the short-chain fibrils also start from extended monomers? This important information is apparently missing.
- 15) On page 5 it is mentioned that "The centroid of the most populous structure is 6.1 angstroms backbone RMSD, so we consider it inaccurate" and later "the MELD predictor does give the correct general locations of the turns and beta-sheets". This seems contradictory, please explain.
- 16) On page 5 in the Limitations sections, it is not clear to me why the authors discuss 5 fibrils (2mxu, 2onv, 3ovl, 4onk, 4 rop) considering that their method didn't fail there. In principle, 2omq and 2ona are the only 2 fibrils that were not well predicted, the first because of sampling and the latter because of the restraint protocol. Did the authors attempted to fix these problems e.g. by simulating longer or changing the restraints?
- 17) On page 7, it is stated that "We have used the final 50 ns of simulation for the analysis of our results" and later that "the first 250 ns of trajectory frames were rejected from clustering" (= the authors used the last 750 ns). Which statement is true?
- 18) Table 1 (page 9). I guess the native contacts, TM-score and GDT_TS refer to the cluster C0 but this information is missing.
- 19) Also, related to Table 1 (and Table 2 as well), the population of C0 is only in some cases 0.4 or more, which is the limiting value to be confident that simulations found the native state (reference 25). Can the authors comment on that?
- 20) Table 2 again, did the authors take the biological assembly PDB file like they did for the proteins mentioned in Table 1? But there is no biological assembly for 2beg.
- 21) Figure captions are extremely brief. I would expand them a bit for clarity.
- 22) Figure 2 (page 12). RMSD definition and units are missing. 2m5n is a long-chain fibril so it should not be here. 6 short chain fibrils are missing.
- 23) Figure 1 and 4. Please check for consistency and decide whether to include angstroms in the figure or not.
- 24) Figure 5 (page 15). RMSD definition and units are missing. It should be stated they used the so-called called S1 restraints only. S2 restraints could be added to 2beg.
- 25) Tables 1 and 2 of the supplementary information. As already mentioned, it would be very helpful to add the presumed oligomer state (tetramer, octamer, etc.) and the experimental method or experimental restraints actually used in the simulations. They should explain whether the number of strands are per monomer or per oligomer.

Reviewer #3 (Remarks to the Author):

The manuscript by Sharma and Dill reports on the use of MELD accelerated molecular dynamics simulations to predict the structures of amyloid fibrils using experimental data. The determination of structures for amyloid fibrils remains an important problem in structural biology. Accurate simulations of these systems is of current interest in the field, and the problem is well motivated in the introduction. MELD is an established method that has been clearly described in past papers from the Dill lab.

There are, however, some major issues with the manuscript in its present form, as described below:

- (1) Insufficient detail in description of methods to allow reproducibility and understanding:
 - The description of "How MELD leverages data" doesn't provide sufficient detail on how MELD works, and does not cite relevant past references from the Dill group.
 - What parameters have been used in the replica exchange simulations? How frequently are

exchanges carried out? How many transits of the T-ladder occurred during the simulations?

- The authors "generate at least 1 microsecond trajectories for all systems". —> Based on the methods description, there is no way to tell how much simulation has been done.

(2) Possible artefacts of parameters:

- The authors have used an implicit solvent model with periodic boundary conditions "to remove edge effects". It's unclear what these edge effects would be for an implicit solvent and why the authors would have used periodic boundary conditions in the first place.

- Similarly, the authors used the Berendsen barostat, which does not produce a physical ensemble.

(3) Figures lack essential information

- Units, axes labels, numbers are missing from **all** figures. It is left to the reader to guess the units.

- Estimates of uncertainty and error are missing from all figures.

(4) Conclusions not sufficiently supported by the data:

- The authors main conclusion, as stated in the last sentence of the discussion: "MELD x MD successfully predicts the relative populations of structures in 17 simulations out of 23" is not supported by the results. What is the criteria for a successful prediction here?

(5) The major advance in this paper is not well-established

- Figure 2 would represent a comparison to other structure predictions — the comparison is not carried out objectively. What is the argument for using MELD x MD when FibPredictor provides a lower RMSD in every case but one? Having a more expensive method that performs less well on average than existing methods is not a clear advance in the field.

- Discussion section is lacking an objective discussion/comparison to existing methods. The discussion section is actually only a few sentences and the conclusion section is missing.

Overall, this paper is poorly written and lacks sufficient information to allow the reader to evaluate or reproduce the authors' work (unlike past papers from this research group on MELD, which is, indeed, a promising simulation method!).

Reviewer 1 : Previously the authors' laboratory has developed an approach that applies Bayesian inference to molecular dynamics simulations which has shown efficacy in the prediction of protein structures using limited and imperfect information. The acceleration this method provides increases the tractability of applying physics-based guided molecular dynamics simulations to problems otherwise typically solved using knowledge-based molecular modelling packages. In this manuscript the authors make use of this technique to build models of amyloid fibrils.

While the amyloidogenic peptides appear to represent a valid target for the MELDxMD method the paper in its present form requires significant changes before it would be suitable for publication. In particular, the application of this technique to the target molecules considered is not sufficiently systematic to really really validate its use. In addition more consideration should be given to the amount of extraneous information provided on the quality of the resulting models. Referencing needs to be improved and key details are absent from the methods section. I provide details below.

Major points:

□-- **R1.1:** For the short-chain fibrils, the amount of information being provided for a relatively simple system is extensive (peptide backbone restraints; inter-strand distances; inter-sheet distances; parallel/anti-parallel arrangement). With such constraints it is unsurprising that the arrangements that emerge are consistent with the PDB structures, and given the significant correlation between side-chain directionality and backbone dihedrals (which have been constrained) it is not especially informative that side-chain orientations are similar between the models and the structures. It is of greater interest as to how the correspondence with the reference structure degrades with the removal of each piece of information for the different peptides. Essentially the purpose of this manuscript is to advocate use of this software in the modelling of fibrils, so a proper go at validating of the minimal requirements in this particular application is crucial.

Us: In our revision (p 12), we now describe such simulations for short fibrils using limited amounts of data, including with no data at all (unguided MELD). However, we also make the point that all the data we use is quite general information and all is derivable from the knowledge of strand arrangements. We have also shown the side-chain orientations of all short-chain fibrils for the general MELD protocol in Fig. 2.

-- **R1.2:** Contacts are designated as Cbeta atoms within 7.5A of one another. Why this definition rather than consideration of the side-chain atoms - the reason should be stated. Contacts are designated as Cbeta atoms within 7.5A of one another. Why this definition rather than consideration of the side-chain atoms - the reason should be stated.

Us: We observed that the qualitative results does not change much by using C β -C β distances, and/or side-chain COM distances, when we are consistent with using a single definition for all structures.

-- **R1.3:** The authors compare with modelling one software application that apparently applies rigid rotation and translation to build fibrils with model optimisation limited to side-chain rotamer packing; and two that are based on docking. There is no mention in the methods or elsewhere of the input parameters used, and it is not explicitly stated whether the same a priori information is used as input - all of these details are essential and must be stated in the methods section. It is surprising that a comparison has not been made with a software that allow similar degrees of freedom of molecular dynamics: a more informative comparison would involve use of molecular modelling software e.g Rosetta or Modeller that is provided with the same input information through the use of dihedral and ambiguous distance constraints.

Us: First, we have now made what comparisons are possible. But, Modeller is not testable here. We contacted Andrej Sali and the Modeller group, but crucially there is no way to use it without homology modeling information, none of which is used here. And, we made a similar attempt with Rosetta. However, none of the public Rosetta software or servers produces any useful structures for short fibrils. Predicting fibrils with Rosetta is still in a research stage, having led to only one publication so far, as far as we know, describing one long fibril, PDB 2MPZ. So, as far as we can tell, there is no other comparable software that uses molecular coordinates. So, second, we have compared our results to the coarser methods, Fibpredictor, Z-Dock and ClusPro, which were previously compared in a horse race among themselves (Reference 33). The value we now add by doing this is not only testing our molecular method, but also broadening to all the 8 classes of amyloid fibrils by each given method. Third, most importantly, the main value of the present work is to introduce a method that can ultimately give the physics, dynamics, free energies and conformational populations in fibrils. The present tests of structural predictions is just a precursor to that goal. Our point here is just to show that our accelerated MD modeling is sufficient to produce amyloid structures comparable to those of the more empirical structures-only methods.

-- **R1.4:** In regards to the identification of polymorphs, is there any independent experimental evidence you can provide for any of these peptides that support the relative populations you are observing? This would give greater confidence that they are not merely computational artefacts. If there genuinely is no such information then you should refer to prior examples where the ensembles identified by the sof

Us: We have added one additional example polymorphic structure for the sequence VQIVYK. The c1 and c12 clusters of MELD simulations correspond to the experimental polymorphs 3olv and 4np8 respectively.

-- **R1.5:** A more systematic approach to address the question posed by the authors "how much information does MELDxMD need" really requires consideration of several structures. n=1 just isn't enough to give confidence that the requirements have truly been tested

Us: Our revised manuscript now describes new simulations with more information for four structures (2beg, 2e8d, 2m5n and 2mxu).

-- **R1.6:** In the limitations section, the standard MD of the 2mxu structure is said to deviate from that seen in the PDB. Were the same dihedral/distance restraints used in the MD simulation as were used with the MELDxMD approach? If not, these cannot really be directly compared

In the limitations section, the standard MD of the 2mxu structure deviates from what's in the PDB. Were the same dihedral/distance restraints used in the MD simulation as were used in MELD x MD approach?

Us: Good point. In the revised manuscript, we have simulated the PDB 2mxu in MELD starting from native fibril structure. We observe that 2mxu deviates by 3.4 Å. So now we have removed that as an example of force-field error.

-- **R1.7:** Referencing is poor in places and needs to be improved

- 14,15,16,17 in paragraph 1 is misleading, these references do not actually provide examples of ssNMR being used to model amyloids. References should be to studies that actually did this.

Us: In the revise manuscript we have added the appropriate references (page 2, reference 14-17).

- 27,28,29,30 on p3 are not examples of simulations leading to kinetically trapped oligomeric structures, I cannot see why they are included here.

Us: We have removed these references now.

- Long enumerated lists of exemplary references to make a single point is lazy referencing and can be substituted with one or two particularly relevant papers instead

Us: Done.

- (18,19,20,21,22,23,24,25) is excessive self-citation and can be trimmed to the most pertinent papers

Us: Done (page3, reference 20-23).

- Look out for any other statements where apparently supporting references are provided that are actually not

Us: Done.

Minor points:

1. The term 'sub-haystacking' on p2 is a very obscure one. This requires some exposition or substitution with another phrase.

Us: Done (page2, last paragraph, line3).

2. It should be explicitly stated in the text and/or supplementary table 1 and table 2 that the information in the column 'strand arrangement' provides information that is used to direct the simulations

Us: *Done (see page 3, last paragraph, line 5).*

3. Supp tables 1 and 2 column 2 heading Name(Å) appears to be a typo

Us: *Å has been removed.*

4. The PDB structure 2kib is the only short peptide dataset that was not solved by X-ray crystallography but by refinement in XPLOR based on SSNMR data. As such is itself a model built from constraints; it could be removed from this test set or this characteristic at least acknowledged

Us: *PDB 2kib has been removed from the test sets of short fibrils.*

4. Provide units on Fig. 2 Y axis

Us: *Done (Fig. 3, page 8).*

5. restraints enforced at 80%... 80% of what? Their magnitude?

Us: *Since experimental data are noisy and ambiguous, therefore we instruct MELD to satisfy only subsets of the data. Restraints enforced at 80% instruct MELD to satisfy 80% of the total data used.*

□ **Reviewer 2:** In this paper by B. Sharma and K.A. Dill, the authors apply Modeling Employing Limited Data accelerated molecular dynamics (MELD x MD) to provide atomistic computational models of a set of amyloid-forming protein fibrils, both “short-chain” (13) and “long chain” (6), using limited experimental information. Moreover, because MELD x MD yields not just single structures but conformational ensembles, the authors claim that their method can also generate polymorphs with similar backbone conformations but distinct side chain packing.

Determining atomic level structures of amyloid fibrils, essential to understand many human diseases, is challenging. Therefore, novel computational approaches, as it is the case of MELD x MD, are welcome. I expect the results of the present work would be beneficial for the community of structural biologists, protein folding and beyond.

In my opinion, the author’s claims are not convincingly supported by their data and the clarity of their explanations could be improved:

Major comments:

-- **R2.1:** The choice of the experimental structures should be better explained.

The known limitations of MELD x MD i.e., short sequence lengths, preference for monomeric and dimeric assemblies, low net charge, high secondary structure content, etc. should be mentioned explicitly and the authors should clearly state whether these limitations of MELD x MD were taken into account for the selection process. For the long chains, the authors mentioned that they were arbitrarily chosen. Is this actually the case or did they try to test different primary structures, oligomeric states and strand arrangements?

Us: Done. The selection of fibrils has been discussed in the Supplementary Information (SI). For long fibrils, we tested with different protocols: a) Starting from same number of random monomers as in the fibrils, or (b) first forming a dimer/trimer, then building up the fibrils. This has been explained in details now in the revised manuscript.

-- **R2.2:** The number of chains in the biological assembly (oligomeric state) is an overlooked restraint.

The authors emphasize that their method can predict accurate structures with limited experimental data (apart from the obvious knowledge about the sequence) but the number of subunits in the fibrils is barely mentioned. I feel the assumed oligomerization is a very important restraint that should be better explained and discussed. I recommend the authors to include the presumed oligomerization state in Tables 1 and 2 (the authors should be aware that they took the biological assembly file from the PDB but the true oligomerization state of a protein is indeed difficult to ascertain and could be different). Moreover, because MELD x MD has so far only been able to deal with monomers or dimers, the authors should explain clearly why now they can model higher order oligomers (tetramers, octamers and so on).

I would remove all mentions in the text along the lines of “using only knowledge of

strand arrangements”, since this is strictly speaking not true. What I mean is to remove words like “only” or “even” that tend to over-emphasize their claims.

***Us:** In Table 1, we now have inserted a column mentioning the number of subunits in the fibrils. For long fibrils, the structures are generated in two ways: (a) Starting from same number of random monomers as in the fibrils, or (b) first forming a trimer, then building up the fibrils. For fibrils bigger than 200 residues, this latter protocol converged much faster. This is explained in detail now; and we have removed ‘only’ and ‘even’.*

-- **R2.3:** The precise experimental data that was used to model the 6 long chain fibrils is not clear.

The structures of the chosen long chain fibrils shown in Table 2 have been solved using different experimental methods (solution NMR, solid-state NMR and others) but it seems that the authors used only NMR distance restraints and backbone restraints. They should specify which and how many restraints per amino acid were employed. Then they could delineate how much information does MELD x MD actually need for predicting long chain fibrils.

Because pdb 2beg was solved by solution NMR, where the number of restraints per amino acids is typically much larger than for solid-state NMR, one would expect a much lower RMSD for 2beg (Figure 5) but this is not the case. Can the authors comment on that? In the main text and in the legend of figure 5, it should be mentioned that for 2beg the authors used solution NMR data.

Related to this issue, are the differences seen in Figure 5 statistically significant? In other words, would the authors recommend to use specific NMR restraints to guide MELD x MD simulations or generic restraints are enough? Such important conclusion of the author’s results is not clearly stated in the text.

***Us:** These are all good points. We now give more detail on SI (Figure 2) on the data used to generate the structures for long chain fibrils; and on page 12 in article file, and in SI (‘MELD x MD simulation with limited information’) on how much information MELD x MD needs. We’ve found that MELD success also depends on the sequence length and oligomeric state of the fibrils, now described more in the Results and Discussion. In case of pdb 2beg, the native fibril structure also deviates by 2.8 Å RMSD in MELD when all the restraints are used. The RMSD difference between the MELD-generated and native structure in MELD is 1.6 Å. This reflects the effects of the force-field in these structure predictions. The virtue of that point is that forcefields are being improved by a large enterprise of researchers, independently of our work on the MELD sampling strategy.*

-- **R2.4:** Polymorphs are only generated for one fibril.

The claim that MELD x MD can generate polymorphs would be more robust if they could repeat their analyses for other fibrils. Then it would also become clearer whether the crystal structure is always the most stable polymorph.

***Us:** We have added one additional example polymorphic structure for the sequence*

VQIVYK. The c1 and c12 clusters of MELD simulations correspond to the experimental polymorphs PDB 3olv and PDB 4np8 respectively (page9).

Minor comments:

5) After reading the introduction, one would erroneously conclude that atomistic MD simulations of protein fibrils have not been attempted. The authors should cite some key references

e.g. <https://doi.org/10.1038/srep33156> and <https://doi.org/10.1021/acs.jpcc.5b11380>.

Us: *Done (reference 18 and 19).*

6) On page 1, the sentence “And, computational methods are often rooted in bioinformatics, ...” is very hard to follow. Please consider rewriting.

Us: *Done. We have mentioned- ‘Some computational methods are rooted in bioinformatics’ (page 2, paragraph 2, line6)*

7) Page 2, Because this work is based on the MELD x MD method, references 18-25, which represent the state-of-the-art of MELD x MD, should be better explained.

Us: *Done (page10, reference 34-37).*

8) Page 2, I believe MELD x MD was employed in CASP11 instead (reference 21).

Us: *MELD was employed in both CASP11 and CASP13 (reference 23,24)*

9) The arbitrary distinction between short- and long-chain fibrils (e.g. 10 residues threshold) should be briefly mentioned.

Us: *Done (page 3, paragraph2, line 2; and Table1)*

10) Please check for consistency between RMSDs definitions, sometimes is Calpha only, others Calpha and Cbeta, backbone, heavy atoms, etc. It is mentioned that the RMSD is calculated relative to the first model in the PDB but sometimes there are 10 experimental models so other approaches (e.g. computing their centroids) might be more accurate. The authors should justify their chosen method.

Us: *For evaluation of prediction quality, we have considered C α RMSD. For short fibril, for the checking the orientation of side-chains (Fig. 2 and Fig.3) we have used all heavy atom RMSD. For consistency of results, we have used MELD structure from top cluster in Fig. 3, whereas previously we used the best MELD structure.*

For structure comparison, we have used ‘Biological assembly’ file from PDB. PDB 2beg does not have Biological assembly file, so used the first model in the PDB as reference.

11) Page 2, please add references for the scoring functions (GDT_TS, Tm-score, etc.)

Us: *Done (reference 26 and 27).*

12) On page 4 it is mentioned that “The side-chain orientations are well predicted from the backbone restraints alone”. How is this quantified?

Us: *In page 7 of the revised manuscript, we have discussed about side-chain orientation.*

13) On page 4, I would move the paragraph starting with “All these methods” to the discussion section since this is more a conclusion than a result.

Us: *This paragraph has been removed.*

14) On page 5 they claim that “MELD X MD successfully predicts the fibril structures starting from extended monomers”. Do the simulations of the short-chain fibrils also start from extended monomers? This important information is apparently missing.

Us: *Yes, for short fibrils, the simulation started from randomly placed extended monomers. This has been mentioned now (page4, first paragraph, line 1).*

15) On page 5 it is mentioned that “The centroid of the most populous structure is 6.1 angstroms backbone RMSD, so we consider it inaccurate” and later “the MELD predictor does give the correct general locations of the turns and beta-sheets”. This seems contradictory, please explain.

Us: *The simulations with limited information have been explained in greater detail in SI, and this statement has been removed.*

16) On page 5 in the Limitations sections, it is not clear to me why the authors discuss 5 fibrils (2mxu, 2onv, 3ovl, 4onk, 4 rop) considering that their method didn’t fail there. In principle, 2omq and 2ona are the only 2 fibrils that were not well predicted, the first because of sampling and the latter because of the restraint protocol. Did the authors attempted to fix these problems e.g. by simulating longer or changing the restraints?

Us: *MELD failures and recovery of the failure structures are explained in details now in page 14 and SI.*

17) On page 7, it is stated that “We have used the final 50 ns of simulation for the analysis of our results” and later that “the first 250 ns of trajectory frames were rejected from clustering” (= the authors used the last 750 ns). Which statement is true?

Us: *This has been corrected now (page16, last paragraph, line3).*

18) Table 1 (page 9). I guess the native contacts, TM-score and GDT_TS refer to the cluster C0 but this information is missing.

Us: *We have added C0 (Table 2, page 6).*

19) Also, related to Table 1 (and Table 2 as well), the population of C0 is only in some cases 0.4 or more, which is the limiting value to be confident that simulations found the native state (reference 25). Can the authors comment on that?

Us: *Large population in MELD foretells success in prediction, and in most cases, population exceeding 40% are successful. However, in many cases, native structures are found even when top MELD cluster is below 40% (Fig.9).*

20) Table 2 again, did the authors take the biological assembly PDB file like they did for the proteins mentioned in Table 1? But there is no biological assembly for 2beg.

Us: *For 2beg we have used the first model of PDB as our reference.*

21) Figure captions are extremely brief. I would expand them a bit for clarity.

Us: *Done.*

22) Figure 2 (page 12). RMSD definition and units are missing. 2m5n is a long-chain fibril so it should not be here. 6 short chain fibrils are missing.

Us: *RMSD definition and units added. 2m5n removed. 6 short-chains are missing because; this figure is adapted from reference 30, where, Z-Dock, ClusPro and Fibpredictor are previously tested against each other.*

23) Figure 1 and 4. Please check for consistency and decide whether to include angstroms in the figure or not.

Us: *Done. We included angstroms.*

24) Figure 5 (page 15). RMSD definition and units are missing. It should be stated they used the so-called called S1 restraints only. S2 restraints could be added to 2beg.

Us: *Figure 5 has been removed.*

25) Tables 1 and 2 of the supplementary information. As already mentioned, it would be very helpful to add the presumed oligomer state (tetramer, octamer, etc.) and the experimental method or experimental restraints actually used in the simulations. They should explain whether the number of strands are per monomer or per oligomer.

Us: *Done. Table 1 and 2 of supplementary information has been shifted to the main text (Table 1).*

Reviewer 3 □: The manuscript by Sharma and Dill reports on the use of MELD accelerated molecular dynamics simulations to predict the structures of amyloid fibrils using experimental data. The determination of structures for amyloid fibrils remains an important problem in structural biology. Accurate simulations of these systems are of current interest in the field, and the problem is well motivated in the introduction. MELD is an established method that has been clearly described in past papers from the Dill lab.

There are, however, some major issues with the manuscript in its present form, as described below:

□-- **R3.1:** Insufficient detail in description of methods to allow reproducibility and understanding:

- The description of “How MELD leverages data” doesn’t provide sufficient detail on how MELD works, and does not cite relevant past references from the Dill group.
- What parameters have been used in the replica exchange simulations? How frequently are exchanges carried out? How many transits of the T-ladder occurred during the simulations?
- The authors “generate at least 1 microsecond trajectories for all systems”. —> Based on the methods description, there is no way to tell how much simulation has been done. The authors should give more detail of “How MELD leverages data”, cite more of their lab's past work, state what parameters were used in the REMD, frequency of exchanges, and describe the total amount of simulation.

Us: These details about MELD method, replica exchange parameters have been included in the Methods section (page15) and SI.

-- **R3.2:** Possible artefacts of parameters:

- The authors have used an implicit solvent model with periodic boundary conditions “to remove edge effects”. It’s unclear what these edge effects would be for an implicit solvent and why the authors would have used periodic boundary conditions in the first place.
- Similarly, the authors used the Berendsen barostat, which does not produce a physical ensemble.

Possible artifacts in parameters section, mention of periodic boundary and Berendsen barostat, which does not produce physical ensemble.

Us: We appreciate the reviewer’s attention to detail. We removed that paragraph from the parameter section.

-- **R3.3:** Figures lack essential information

- Units, axes labels, numbers are missing from *all* figures. It is left to the reader to guess the units.
 - Estimates of uncertainty and error are missing from all figures.
- Figures are missing units and axis labels and estimates of uncertainty.

Us: *Done.*

-- **R3.4:** Conclusions not sufficiently supported by the data:

- The authors main conclusion, as stated in the last sentence of the discussion: “MELD x MD successfully predicts the relative populations of structures in 17 simulations out of 23” is not supported by the results. What is the criteria for a successful prediction here? Conclusions not sufficiently supported by the data.

Us: *Done. Figure 9 shows the relative populations of MELD successes and MELD failures (page14).*

-- **R3.5:** The major advance in this paper is not well-established

- Figure 2 would represent a comparison to other structure predictions — the comparison is not carried out objectively. What is the argument for using MELD x MD when FibPredictor provides a lower RMSD in every case but one? Having a more expensive method that performs less well on average than existing methods is not a clear advance in the field.

- Discussion section is lacking an objective discussion/comparison to existing methods. The discussion section is actually only a few sentences and the conclusion section is missing.

Us: *Same response as R1.3, about FibPredictor.*

-Instead of Discussion section, now we have 'Results and Discussion'. Conclusion section is added.

Reviewers' comments:

Reviewer #1 (Remarks to the Author):

This manuscript concerns the use of the MELD approach, developed in the authors' laboratory, to guide and accelerate molecular dynamics simulations of amyloid fibril assemblies. The development of a new approach to increase the tractability of atomistic treatment of these molecules is a worthwhile aim.

The revised manuscript incorporates a significant number of changes, including additional data in both the main document and as supplementary information. Several less-well or unsupported claims have been supported by new data, removed, or clarified, and referencing has been improved. Crucially, the new data included provides an improved basis on which a reader can evaluate the claims of the study. This has resulted overall in a higher-quality submission.

Major comments:

My main remaining issue with this study was alluded to by some of my previous comments, which I will attempt to re-articulate: in their enthusiasm for the technique they are applying, there are a couple of points in the text where I am concerned the authors have not taken care to give sufficient acknowledgement/appraisal of the extent of restraints operating on the system during the simulations. In Figure 1, the legend states "Input guidance to MELD was only whether strands are parallel or antiparallel". However from the text at the bottom of page 3, they have: (i) imposed dihedral restraints ensuring residues adopt a beta-strand conformation; (ii) imposed an inter-strand distance; (iii) imposed an inter-sheet distance; (iv) supplementary Fig. 1 implies the inter-strand distances are between specific residues encouraging or even ensuring in-register alignment of the beta-strands and also that (v) the inter-sheet restraints are between centres of mass?/central residues? of the strands that themselves impose a directionality to sheet-sheet interfaces; (vi) presumably the dihedral restraints and inter-strand distance restraints are applied to all residues although this is not stated anywhere. I cannot overemphasise the importance of including details such as (iv), (v), (vi) (if correct) explicitly and clearly in the text and these must not be glibly ignored in summative statements such as the one made in Figure 1. Of course if my inferences for these points are incorrect, whatever accurately *and completely* describes what was done for the short fibrils must be stated, including any further information. In the methods section, the statement "Our typical inputs to MELD are (i) initial configuration of peptide that are randomly placed, (ii) externally supplied distance and dihedral angle restraints, and (iii) Secondary structures information (if available)" provides no further guidance either.

Correspondingly, the statement in the abstract "Ten short-chain fibril structures are accurately predicted using no experimental data except knowledge of strand directions" is over-simplistic and potentially misleading, and as such should also be clarified.

In a related vein is the analysis on page 7, which disregards the fact that side-chain orientations are not independent of backbone dihedrals and will be constrained by the restrained inter-strand distances. I raised this point in my previous review. The indirect influence of these restraints are overlooked by the statement "even though we have not restrained the side chain atoms in the MELD simulation" at the end of para 1 and "without requiring any such directive restraints". It doesn't detract from the MELD approach but a less sales-y, finessed treatment is appropriate here.

Minor comments:

I appreciate the inclusion of the supplementary ODS data file, it should however be annotated for it to be properly of use, including table headings on all sheets, units, figure cross-references, brief descriptions, etc.

The manuscript and supplementary data are peppered with spurious punctuation, grammatical errors, inappropriate capitalisation, and missing spaces; please proofread carefully.

Reviewer #2 (Remarks to the Author):

The revised manuscript by Sharma & Dill has improved significantly. Also, the answers to my concerns are very satisfactory. I appreciate that the authors made a great effort to improve the quality of their presentation (particularly the figures) and to support their claims with more computational details and convincing results. The authors now clearly discuss i) the prediction of side-chain orientations and polymorphs, ii) the critical information that MELD x MD needs as input in order to predict amyloid fibril structures, iii) as well as the limitations of their method and how to overcome them.

There are still a few points that the authors may want to clarify:

- 1) Page 3 last line of the introduction, and conclusions chapter on page 15. I would specify that MELD predicts 5 out of 6 longer-chain amyloid fibrils.
- 2) Figure 1 (page 5). I suggest making the caption much clearer like "Computationally predicted vs. experimentally determined native structures of 12 short fibrils". Generally speaking, I would avoid the use of the word "true" when the authors are referring to a structure deposited in the PDB database. I guess the authors actually mean "experimentally determined structures" or "structures based on experimental evidence" (e.g. page 14).
- 3) While in general it is a good idea to use the PDB code to identify the fibrils, in the case of polymorphs this is quite confusing: The sequences have polymorphs, not the PDB files! In particular, the second paragraph of page 9 would be better understood as "Fig. 4 shows the polymorphic structures generated in MELD simulations for VQIVYK and NNQNTF sequences". In addition, what is the reason the authors compare their simulated fibrils of VQIVYK with only two polymorphs? At least 4 polymorphs of this sequence can be found in the PDB (3ovl, 4np8, 2on9 and 5k7n).
- 4) Just a suggestion. It would perhaps be more intuitive to name SO, S1, S2 and S3 to the restraints applied to short fibrils, and LO, L1, L2 and L3 to the restraints on long fibrils.
- 5) Table 1 of SI (page 2). I encourage the authors to add the UniprotKB code to the 2 tables in order to unambiguously identify the sequence of the fibrils.
- 6) Page 17 of SI. This text has been largely duplicated from the main text and could be removed.
- 7) Thorough proofreading would remove the missing typos e.g. "well-separated" (page 4), "structures" (page 12), "the method is expected to" (page 15).

Reviewer 1: This manuscript concerns the use of the MELD approach, developed in the authors' laboratory, to guide and accelerate molecular dynamics simulations of amyloid fibril assemblies. The development of a new approach to increase the tractability of atomistic treatment of these molecules is a worthwhile aim.

The revised manuscript incorporates a significant number of changes, including additional data in both the main document and as supplementary information. Several less-well or unsupported claims have been supported by new data, removed, or clarified, and referencing has been improved. Crucially, the new data included provides an improved basis on which a reader can evaluate the claims of the study. This has resulted overall in a higher-quality submission.

Major comments:

My main remaining issue with this study was alluded to by some of my previous comments, which I will attempt to re-articulate: in their enthusiasm for the technique they are applying, there are a couple of points in the text where I am concerned the authors have not taken care to give sufficient acknowledgement/appraisal of the extent of restraints operating on the system during the simulations. In Figure 1, the legend states "Input guidance to MELD was only whether strands are parallel or antiparallel". However from the text at the bottom of page 3, they have: (i) imposed dihedral restraints ensuring residues adopt a beta-strand conformation; (ii) imposed an inter-strand distance; (iii) imposed an inter-sheet distance; (iv) supplementary Fig. 1 implies the inter-strand distances are between specific residues encouraging or even ensuring in-register alignment of the beta-strands and also that (v) the inter-sheet restraints are between centres of mass?/central residues? of the strands that themselves impose a directionality to sheet-sheet interfaces; (vi) presumably the dihedral restraints and inter-strand distance restraints are applied to all residues although this is not stated anywhere. I cannot overemphasise the importance of including details such as (iv), (v), (vi) (if correct) explicitly and clearly in the text and these must not be glibly ignored in summative statements such as the one made in Figure 1. Of course if my inferences for these points are incorrect, whatever accurately *and completely* describes what was done for the short fibrils must be stated, including any further information. In the methods section, the statement "Our typical inputs to MELD are (i) initial configuration of peptide that are randomly placed, (ii) externally supplied distance and dihedral angle restraints, and (iii) Secondary structures information (if available)" provides no further guidance either.

Us: In our revision (p 3, highlighted in blue color), we now explain the restraints used for short fibrils in more detail. The legend from Figure 1 has been changed. The restraints information used in MELD for inter-strand restraints are derived from the external input knowledge of strand-arrangements of peptides in fibrils. We have incorporated inter-strand distance restraints and dihedral angle restraints to all

corresponding residues in short fibrils. However, for inter-sheet restraints, we have used a general protocol for all structures. We have applied inter-sheet restraints to three central residues of each strand, among one another (see Supplementary Figure 1, which has been changed for clarity). Since this experimental information is ambiguous and uncertain, therefore we have enforced the restraints at a level of 80% accuracy.

2) Correspondingly, the statement in the abstract “Ten short-chain fibril structures are accurately predicted using no experimental data except knowledge of strand directions” is over-simplistic and potentially misleading, and as such should also be clarified.

Us: *Done (page 1, first paragraph).*

3) In a related vein is the analysis on page 7, which disregards the fact that side-chain orientations are not independent of backbone dihedrals and will be constrained by the restrained inter-strand distances. I raised this point in my previous review. The indirect influence of these restraints are overlooked by the statement “even though we have not restrained the side chain atoms in the MELD simulation” at the end of para 1 and “without requiring any such directive restraints”. It doesn’t detract from the MELD approach but a less sales-y, finessed treatment is appropriate here.

Us: *Done (page 7, first paragraph, last sentence).*

Minor comments:

1) I appreciate the inclusion of the supplementary ODS data file, it should however be annotated for it to be properly of use, including table headings on all sheets, units, figure cross-references, brief descriptions, etc.

Us: *Done.*

2) The manuscript and supplementary data are peppered with spurious punctuation, grammatical errors, inappropriate capitalisation, and missing spaces; please proofread carefully.

Us: *Done.*

Reviewer 2: The revised manuscript by Sharma & Dill has improved significantly. Also, the answers to my concerns are very satisfactory. I appreciate that the authors made a great effort to improve the quality of their presentation (particularly the figures) and to support their claims with more computational details and convincing results. The authors now clearly discuss i) the prediction of side-chain orientations and polymorphs, ii) the critical information that MELD x MD needs as input in order to predict amyloid fibril structures, iii) as well as the limitations of their method and how to overcome them.

There are still a few points that the authors may want to clarify:

1) Page 3 last line of the introduction, and conclusions chapter on page 15. I would specify that MELD predicts 5 out of 6 longer-chain amyloid fibrils.

Us: *Done.*

2) Figure 1 (page 5). I suggest making the caption much clearer like “Computationally predicted vs. experimentally determined native structures of 12 short fibrils”. Generally speaking, I would avoid the use of the word “true” when the authors are referring to a structure deposited in the PDB database. I guess the authors actually mean “experimentally determined structures” or “structures based on experimental evidence” (e.g. page 14).

Us: *Done page (5, and page 15).*

3) While in general it is a good idea to use the PDB code to identify the fibrils, in the case of polymorphs this is quite confusing: The sequences have polymorphs, not the PDB files! In particular, the second paragraph of page 9 would be better understood as “Fig. 4 shows the polymorphic structures generated in MELD simulations for VQIVYK and NNQNTF sequences”. In addition, what is the reason the authors compare their simulated fibrils of VQIVYK with only two polymorphs? At least 4 polymorphs of this sequence can be found in the PDB (3ovl, 4np8, 2on9 and 5k7n).

Us: *In our revision, we have used the sequence instead of the PDB code (page 10).*

Some polymorphic structures were excluded initially because of different numbers of subunits (or oligomerization states) in the crystal structures. We have now added additional polymorphs of the VQIVYK sequence PDB 2on9 in our analysis. However, the crystal structure of PDB 5k7n has only a single parallel β -sheet, so it is not incorporated in our study. The same is true for sequence KLVFFA, which has three experimental polymorphic structures (3ow9, 2y2a and 2y29). However, PDB 2y2a and 2y29 sequences have only a single chain in the Biological Assembly file in the PDB.

4) Just a suggestion. It would perhaps be more intuitive to name SO, S1, S2 and S3 to the restraints applied to short fibrils, and LO, L1, L2 and L3 to the restraints on long fibrils.

Us: *Done.*

5) Table 1 of SI (page 2). I encourage the authors to add the UniprotKB code to the 2 tables in order to unambiguously identify the sequence of the fibrils.

Us: *Done. We could not find UniProtKB code for PDB 2ona, 2onv and 4onk. So they are omitted.*

6) Page 17 of SI. This text has been largely duplicated from the main text and could be removed.

Us: *Done.*

7) Thorough proofreading would remove the missing typos e.g. “well-separated” (page 4), “structures” (page 12), “the method is expected to” (page 15).

Us: *Done.*

REVIEWERS' COMMENTS:

Reviewer #1 (Remarks to the Author):

In the previous versions of the manuscript, my main concern has been inadequate detail/acknowledgement of extrinsic and implicit restraints and the extent to which these have influenced the progression and outcome of the simulations. The second version was a substantial improvement on the first; with the further changes made here, there is now I believe sufficient information for a reader to evaluate the method used and the claims made by the authors.

Final minor points:

- RMSD values differ in Table 2, Figure 2 and Figure 4 because they have been defined in different ways. For clarity please specify in Figure/Table legends ("backbone RMSD", "heavy-atom RMSD" etc)
- On p15 "adding a few inter-monomer beta-sheet restraints" ... "some correct inter-sheet restraints": please give details of these in the Supp Fig 13 legends, either how many were introduced and how these were selected or the actual specifications of the restraints themselves. Otherwise it is not clear how you rescued them.

Suggestions only:

- Table 1 in Supp materials is very informative, consider including its details in the main text by merging with Table 1 (eg removing "monomer sequence length" - evident from the sequence anyway
- abbreviating parallel/anti-parallel to "P", "AP")
- Consider reformatting Fig. 4 to make it more consistent with the styling (font etc) of the other figures
- Double-check text again for consistent use of past/present tense and also typos/grammar, these are just a few I happened to make a note of:
 - p8 para2 l5 "While Fibpredictor" -> delete "While"
 - p8 para2 l8 "calculated for of all" -> delete "of"
 - p10 end para 1 "PBD" -> "PDB"
 - p10 para 4 "took longer time" -> delete "time"
 - p10 para 4 Figure 6 referenced before Figure 5
 - p12 "the cutoff for folded to native fibril" -> "the cutoff used to define a native fibril" (there are a few other sentences like this as well)
 - p14 l3 "however, limited amount of" -> "however, a limited number of"
 - p17 l~12 "restrains" -> "restraints"
 - p17 l~30 "restraints information are" -> "restraints are"

Reviewer 1:

In the previous versions of the manuscript, my main concern has been inadequate detail/acknowledgement of extrinsic and implicit restraints and the extent to which these have influenced the progression and outcome of the simulations. The second version was a substantial improvement on the first; with the further changes made here, there is now I believe sufficient information for a reader to evaluate the method used and the claims made by the authors.

Final minor points:

1) RMSD values differ in Table 2, Figure 2 and Figure 4 because they have been defined in different ways. For clarity please specify in Figure/Table legends ("backbone RMSD", "heavy-atom RMSD" etc)

Us: *Done.*

2) On p15 "adding a few inter-monomer beta-sheet restraints" ... "some correct inter-sheet restraints": please give details of these in the Supp Fig 13 legends, either how many were introduced and how these were selected or the actual specifications of the restraints themselves. Otherwise it is not clear how you rescued them.

Us: *Done.*

Suggestions only:

1) Table 1 in Supp materials is very informative, consider including its details in the main text by merging with Table 1 (eg removing "monomer sequence length" - evident from the sequence anyway - abbreviating parallel/anti-parallel to "P", "AP")

Us: *Instead of merging the Supplementary Table 1 with main text Table 1, we have added two more column in Table 1 of main text (Name and sequence) to make it more informative. The Supplementary Table 1 contains some extra information like UniprotKB code (suggested by Reviewer 2 in the last revision) and Experiments and Computations used to solve the fibril structures.*

2) Consider reformatting Fig. 4 to make it more consistent with the styling (font etc) of the other figures

Us: *Done.*

3) Double-check text again for consistent use of past/present tense and also typos/grammar, these are just a few I happened to make a note of:

- p8 para2 15 "While Fibpredictor" -> delete "While"
- p8 para2 18 "calculated for of all" -> delete "of"
- p10 end para 1 "PBD" -> "PDB"
- p10 para 4 "took longer time" -> delete "time"
- p10 para 4 Figure 6 referenced before Figure 5
- p12 "the cutoff for folded to native fibril" -> "the cutoff used to define a native fibril" (there are a few other sentences like this as well)
- p14 l3 "however, limited amount of" -> "however, a limited number of"
- p17 l~12 "restrains" -> "restraints"
- p17 l~30 "restraints information are" -> "restraints are"

Us: *Done.*